

# Effects of bamlanivimab alone or in combination with etesevimab on subsequent hospitalization and mortality in outpatients with COVID-19: a systematic review and meta-analysis

Yu-Lin Tai[1,2,*], Ming-Dar Lee[1,2,*], Hsin Chi[3,4], Nan-Chang Chiu[3,4], Wei-Te Lei[1,2,5], Shun-Long Weng[3,6], Lawrence Yu-Min Liu[3,7], Chung-Chu Chen[7,8], Shih-Yu Huang[7], Ya-Ning Huang[1,2] and Chien-Yu Lin[1,2,3]

[1] Pediatrics, Hsinchu MacKay Memorial Hospital, Hsinchu, Taiwan
[2] Pediatrics, Hsinchu Municipal MacKay Children's Hospital, Hsinchu, Taiwan
[3] Medicine, MacKay Medical College, New Taipei, Taiwan
[4] Pediatrics, MacKay Children's Hospital, Taipei, Taiwan
[5] Graduate Institute of Clinical Medical Sciences, College of Medicine, Chang Gung University, Taoyuan, Taiwan
[6] Department of Obstetrics and Gynecology, Hsinchu MacKay Memorial Hospital, Hsinchu, Taiwan
[7] Department of Internal Medicine, Hsinchu MacKay Memorial Hospital, Hsinchu, Taiwan
[8] Teaching Center of Natural Science, Minghsin University of Science and Technology, Hsinchu, Taiwan
* These authors contributed equally to this work.

Corresponding author
Chien-Yu Lin,
mmhped.lin@gmail.com

## ABSTRACT

**Background:** Coronavirus disease 2019 (COVID-19) has caused an enormous loss of life worldwide. The spike protein of the severe acute respiratory syndrome coronavirus 2 is the cause of its virulence. Bamlanivimab, a recombinant monoclonal antibody, has been used alone or in combination with etesevimab to provide passive immunity and improve clinical outcomes. A systematic review and meta-analysis was conducted to investigate the therapeutic effects of bamlanivimab with or without etesevimab (BAM/ETE) treatment.

**Methods:** Our study was registered in PROSPERO (registry number CRD42021270206). We searched the following electronic databases, without language restrictions, until January 2023: PubMed, Embase, medRxiv, and the Cochrane database. A systematic review and meta-analysis was conducted based on the search results.

**Results:** Eighteen publications with a total of 28,577 patients were identified. Non-hospitalized patients given bamlanivimab with or without etesevimab had a significantly lower risk of subsequent hospitalization (18 trials, odds ratio (OR): 0.37, 95% confidence interval (CI): [0.29–0.49], $I^2$: 69%; $p < 0.01$) and mortality (15 trials, OR: 0.27, 95% CI [0.17–0.43], $I^2$: 0%; $p = 0.85$). Bamlanivimab monotherapy also reduced the subsequent risk of hospitalization (16 trials, OR: 0.43, 95% CI [0.34–0.54], $I^2$: 57%; $p = 0.01$) and mortality (14 trials, OR: 0.28, 95% CI [0.17–0.46], $I^2$: 0%; $p = 0.9$). Adverse events from these medications were uncommon and tolerable.

**Conclusions:** In this meta-analysis, we found the use of bamlanivimab with or without etesevimab contributed to a significantly-reduced risk of subsequent hospitalization and mortality in non-hospitalized COVID-19 patients. However, resistance to monoclonal antibodies was observed in COVID-19 variants, resulting in the halting of the clinical use of BAM/ETE. Clinicians' experiences with BAM/ETE indicate the importance of genomic surveillance. BAM/ETE may be repurposed as a potential component of a cocktail regimen in treating future COVID variants.

# INTRODUCTION

The crisis caused by the coronavirus disease 2019 (COVID-19), fueled by severe acute respiratory syndrome coronavirus 2 (SARS-CoV-2), continues to be a substantial health threat. As of July 31, 2021, more than 200 million patients have been infected, resulting in more than 4 million deaths (*Chang et al., 2020*; *Lai et al., 2020a*; *Johns Hopkins University & Medicine, 2021*). The overall fatality rate of the initial COVID-19 variants is approximately 2–3%, however, this percentage decreased with the Omicron variants (*Mathieu et al., 2020*; *Finelli et al., 2021*). Those who are elderly, obese, or have underlying systemic diseases have a higher risk for unfavorable outcomes (*Mathieu et al., 2020*; *Lai et al., 2020b*; *Yuki, Fujiogi & Koutsogiannaki, 2020*). The pathophysiology of COVID-19 involves different phases of infection. The direct viral invasion is responsible for the initial injury and immune-mediated inflammatory processes result in subsequent damage to the host. The timely diagnosis and proper management of the disease, such as early quarantine, treatment, and prognosis, contributes to successful patient outcomes (*Gandhi, Lynch & del Rio, 2020*). Several pharmacotherapeutic medications, which use different mechanisms for treatment, have been used to combat COVID-19 in patients with varying disease severity and at different phases of the disease (*Gandhi, Lynch & del Rio, 2020*; *Siemieniuk et al., 2020*; *Yang et al., 2021*). Some antivirals, monoclonal antibodies, and convalescent plasma have been used to target viruses in their initial phase. Remdesivir is just such a medication, and it has been used to inhibit RNA-dependent RNA polymerase (*Moreno et al., 2022*). Some drugs are applied for their immune-modulatory effects in the inflammatory phase, including corticosteroids, janus kinase (JAK) inhibitor, and interleukin-6 (IL-6) inhibitors (*Gautret et al., 2020*; *Lan et al., 2020*). For patients with moderate to severe infections, monoclonal antibody treatments are ineffective, while IL-6 inhibitors, such as tocilizumab, are the mainstay treatment and contribute to better outcomes (*Hariyanto, Hardyson & Kurniawan, 2021*; *Rosas et al., 2021*). However, the effects of treatment are largely unclear and the optimal course of treatment is still under investigation.

COVID-19 has proven to be a highly contagious virus with moderate severity. The severity of Omicron infections was relatively mild, and most people were vaccinated when it appeared. Therefore, many countries adopted a co-existing policy, which allowed populations to return to a more normal life with eased social restrictions. However, the

rapid increase in patients with COVID-19 caused a shortage of medical resources to the point where a medical system collapse could occur. Outpatient treatments with reduced subsequent hospitalization may help to preserve medical resources, thus, monoclonal antibodies were developed to ensure adequate and timely outpatient treatment. SARS-CoV-2 is composed by four major structural proteins including spike, envelope, membrane, and nucleocapsid proteins. Bamlanivimab (BAM) (Eli Lilly, former name: LY-CoV-555) is a type of neutralizing monoclonal antibody derived from the plasma of recovered patients (*Cohen, 2021*; *Ganesh et al., 2021a*). It binds to the epitopes in the receptor binding domain of the spike protein and provides rapid and passive immunity against SARS-CoV-2 (*Jones et al., 2020*; *Taylor et al., 2021*; *Tuccori et al., 2021*). Early monoclonal antibody treatment in patients with mild COVID-19 infection may block the viral replication and improve subsequent clinical outcomes. Etesevimab (ETE) (Eli Lilly, former name: LY-CoV-016) is another monoclonal antibody targeting a different epitope of the receptor binding domain. BAM alone or in combination with ETE (BAM/ETE) was emergency authorized to treat COVID-19 patients by the US Food and Drug Administration in November 2020. For non-hospitalized COVID-19 patients, BAM/ETE treatment decreased the risk of subsequent hospitalization and mortality (*Cohen, 2021*). A single infusion in an outpatient setting decreased the risk of subsequent hospitalization and helped to preserve medical capacity. However, this benefit was not observed in hospitalized patients. Therefore, our team conducted this systematic review and meta-analysis to investigate the effects of BAM ± ETE on patients with COVID-19.

## MATERIALS AND METHODS

### Study design and literature search

We followed the preferred reporting protocol for systematic reviews and meta-analyses guidelines and registered our trial in PROSPERO (registry number CRD42021270206) (*Chi et al., 2021*; *Page et al., 2021*). The written consent of the subjects was waived for the design of a systematic review without identifiable personal data. Comprehensive keywords were applied, including "COVID-19," "severe acute respiratory syndrome coronavirus 2," "SARS-CoV-2," "bamlanivimab," "etesevimab," and "monoclonal antibody" with Boolean operators and medical subject heading terms. The PICOT used in our systematic review was as follows: in COVID-19 patients with mild or moderate severity (P), how monoclonal antibody treatment with BAM ± ETE (I) compared with standard of care (C) affects subsequent risk of hospitalization or mortality (O)? The complete search strategy is shown in File S1. Electronic medical databases were searched from inception to July 31, 2021, including PubMed/Medline, EMBASE, the Cochrane database, and the medRxiv database. Two authors (YL Tai and CY Lin) performed the literature research independently and disagreements were resolved through discussions with the third author (MD Lee).

No constraints were placed on language or the year of publication in order to ensure a comprehensive search and to identify the maximum number of potential articles.

An updated search was performed on 13 Jan 2023 to identify new resources and to evaluate the impact of the new studies.

### Study selection, data extraction, and quality assessment

We included randomized-controlled studies, cohort studies, cross-sectional studies, case control studies, or matched cohort studies to investigate the effects of BAM/ETE on COVID-19 patients. The exclusion criteria were as follows: duplicate publications in different databases, irrelevant articles, animal studies, studies that the infection status was not clearly confirmed, studies with no evaluation of clinical outcomes, simple case reports, studies investigating specific group, prophylactic use of BAM/ETE, single arm studies, editorials, and review articles. Articles were screened using the title and abstract. The effects of BAM/ETE on subsequent hospitalization were the primary measured outcomes. The secondary outcomes were the effects on mortality and adverse events.

Two authors independently appraised the selected articles and performed a quality assessment. The revised Cochrane risk-of-bias tool for randomized trials (RoB 2) was used to assess the randomized-controlled trials and the Newcastle-Ottawa scale for observational cohort studies (*Wells et al., 2021*) were employed. Quality assessments were performed based on selection, ascertainment, causality, and reporting. If the two authors disagreed, a consensus was reached through a discussion with the third author. The following data was obtained from the included studies: name of the first author, study country, study period, participant population, demographic data, dosage of BAM/ETE, clinical outcomes, adverse events, and conclusion. A subgroup analysis of BAM monotherapy and combination therapy was also performed. The prevalence of COVID variants may have affected the pharmacological effects, therefore, the prevalence of variants during the study period was also explored (*Hodcroft, 2021*).

### Statistical analyses

The reported odds ratios (OR) of the included studies were pooled to calculate the OR of BAM/ETE treatment on subsequent hospitalization, mortality, and adverse events. If a meta-analysis was performed, a random-effect regression model was used, assuming that the true effect size was not the same. Heterogeneity was quantified using the Cochran's Q test and $I^2$ statistics (*Higgins et al., 2003*). The heterogeneity was considered to be low, moderate, and high at $I^2 < 50\%$, 50–75%, and >75%, respectively. Sensitivity analysis was conducted to investigate the effects of an individual study by removing it from consideration. Potential publication study bias was evaluated using funnel plots and Egger's regression testing. A *p* value less than 0.05 was considered statistically significant. R software version 4.0.3 (R Foundation for Statistical Computing, Vienna, Austria) was applied for statistical analyses.

## RESULTS

### Included studies and demographic characteristics

A total of 14 studies investigating BAM/ETE were identified in the literature by July 31, 2021. These studies included 28,984 total participants (Fig. 1, Table 1) (*Alam et al., 2021*; *Bariola et al., 2021*; *Chen et al., 2021*; *Corwin et al., 2021*; *Destache et al., 2021*; *Dougan et al., 2021*; *Ganesh et al., 2021b*; *Gottlieb et al., 2021*; *Karr et al., 2021*; *Koehler et al., 2021*; *Kumar et al., 2021*; *Lundgren et al., 2021*; *Piccicacco et al., 2021*; *Webb et al., 2021*). Three
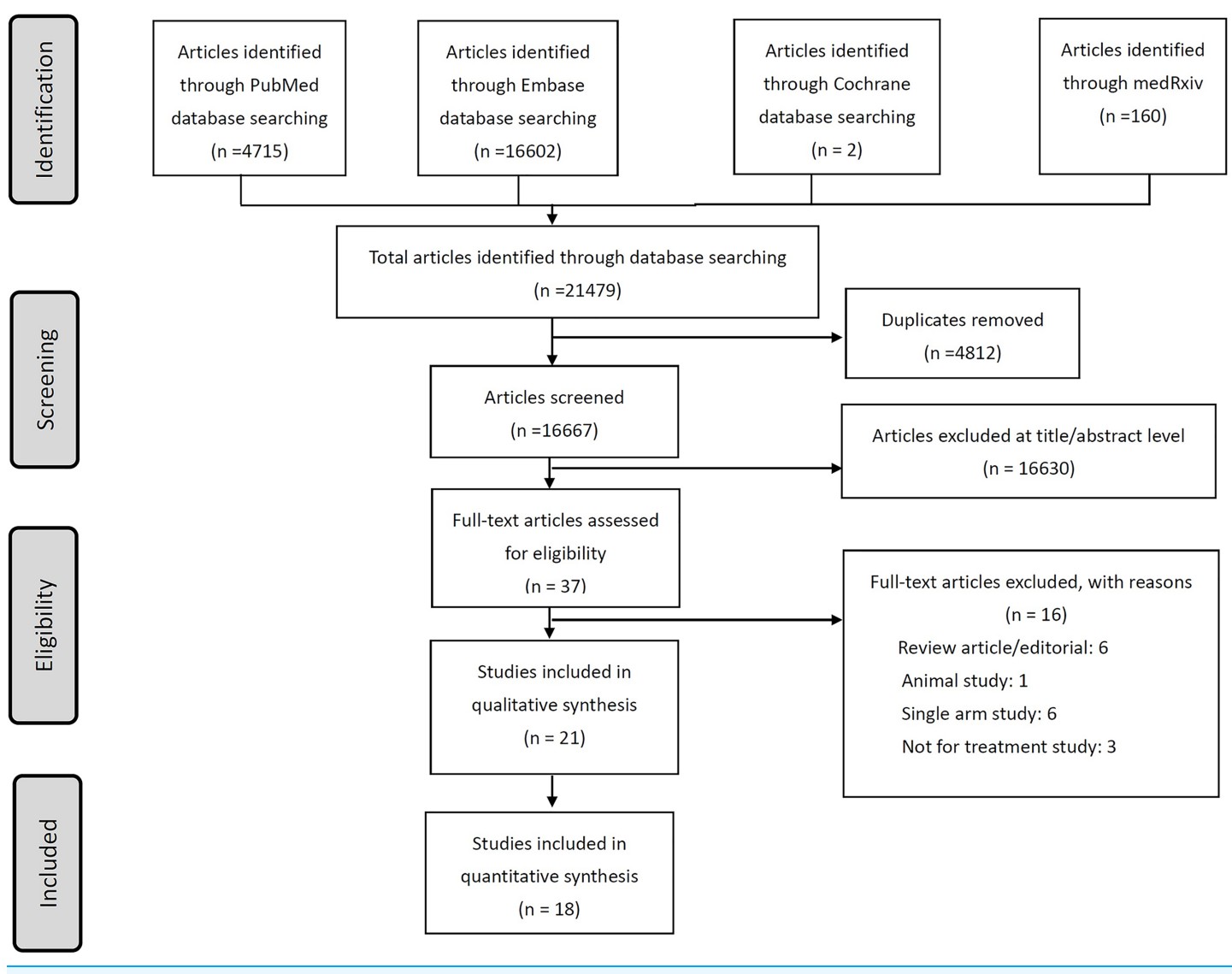

**Figure 1 The flow chart of literature search and review of included studies.**

publications reported on BLAZE-1 phase 2 and phase 3 studies and some participants were found to be overlapping (*Chen et al., 2021*; *Dougan et al., 2021*; *Gottlieb et al., 2021*). One study was conducted in Germany, one in Denmark, and one in Singapore, with the majority of the studies being conducted in the United States. Four studies were randomized-controlled trials and the others were observational cohort studies. Two studies involved hospitalized patients, while the remaining studies focused on non-hospitalized patients (*Koehler et al., 2021*; *Lundgren et al., 2021*). Monotherapy with BAM was used in most of the studies and combination therapy with ETE was investigated in only three studies. The COVID variants of interest in the studies included 21C (Epsilon), 20E (EU1), and B.1.1.7 (Alpha). Seven additional studies were identified in January 2023, and these were included in our analysis (*Chilimuri et al., 2022*; *Cooper et al.,*

**Table 1  Clinical characteristics and demographic data of included studies.**

| Study | Country | Study period (prevalent strain) | Study design | Severity | Participants | Gender, female | Age (years) | Intervention | Control | Outcomes | Safety outcome |
|---|---|---|---|---|---|---|---|---|---|---|---|
| BLAZE-1, monotherapy | USA | June 2020–August 2020 (non-variant) | Phase 2, randomized, double-blind, placebo-controlled | Outpatients, mild/moderate | 452 | 249 | 45 | LY-CoV555, 700, 2,800, 7,000 mg for single dose | Placebo | Virological endpoints, clinical outcomes | Adverse events |
| BLAZE-1, phase 3 | USA | September 2020–December 2020 (21C, Epsilon) | Phase 3, randomized, double-blind, placebo-controlled | >= 12 y/o outpatients, mild/moderate | 1,035 | 538 | 53.8 | 2,800 mg of bamlanivimab and 2,800 mg of etesevimab | Placebo | Virological endpoints, clinical outcomes | Adverse events |
| BLAZE-1, combination | USA | June 2020–September 2020 (non-variant) | Phase 2/3, randomized, double-blind, placebo-controlled | Outpatients, mild/moderate | 577 | 315 | 44.7 | Bamlanivimab (700, 2,800, 7,000 mg), the combination treatment (2,800 mg of bamlanivimab and 2,800 mg of etesevimab) | Placebo | Virological endpoints, clinical outcomes | Adverse events |
| ACTIV-3 | USA, Denmark, Singapore | August 2020–October 2020 (non-variant and 21C) | Randomized, double-blind, placebo-controlled | Hospitalized patients without end-organ failure | 314 | 137 | 61 | LY-CoV555 7,000 mg for single dose | Placebo | Pulmonary ordinal outcome, Time to sustained recovery, Time to hospital discharge | Infusion reaction, Composite safety outcome, Composite safety outcome, organ dysfunction, or serious Coinfection, Death |
| Alam | USA | November 2020–January 2021 (21C, Epsilon) | Retrospective case-control study | Long-term care facility, mild/moderate | 246 | 129 | 82.4 | LY-CoV555 7,000 mg for single dose | Not treated | Clinical outcomes (death and hospitalization) | 4.37% side effects |
| Bariola | USA | December 2020–March 2021 (21C Epsilon and Alpha) | Propensity-matched observational study | Non-hospitalized, mild/moderate | 1,392 | 650 | 67 | 700 mg of bamlanivimab | Not treated | 1. 28-day hospitalization, all-cause mortality 2. Hospitalization or ER without hospitalization | |
| Chilimuri | USA | November 2020–Mar 2021 (21C Epsilon and Alpha) | Retrospective observational study | Mild to moderate | 38 | 22 | 65 | 700 mg of bamlanivimab or casirivimab/imdevimab | Not treated | 30-day hospitalization and mortality | |
| Cooper | USA | November 2020–May 2021 (21C Epsilon and Alpha) | Propensity-matched observational cohort study | Outpatients | 5,758 | 3,159 | 60 | Bamlanivimab, bamlanivimab-etesevimab, casirivimab/imdevimab | Not treated | 1. 14- and 28-day hospitalization 2. Intensive care unit admission, mortality | |

| Study | Country | Study period (prevalent strain) | Study design | Severity | Participants | Gender, female | Age (years) | Intervention | Control | Outcomes | Safety outcome |
|---|---|---|---|---|---|---|---|---|---|---|---|
| Corwin | USA | November 2020–January 2021 (21C) | Cohort study | Outpatients who have BMI ≥35 and/or are age ≥65 years old, mild/moderate | 6,117 | 3,456 | 62.6, 56.7 | 700 mg of bamlanivimab | Not treated | Admission, emergency department visit, death | 5.4% adverse effects |
| Destache | USA | November 2020–December 2020 (21C) | Propensity-matched retrospective cohort study | Outpatients, mild/moderate | 234 | 124 | 72 | 700 mg of bamlanivimab | Not treated | Clinical outcomes | |
| Ganesh | USA | November 2020–February 2021 (21C, Alpha) | Propensity-matched retrospective cohort study l | Outpatients, mild/moderate | 4,670 | 2,306 | 63 | 700 mg of bamlanivimab | Not treated | Clinical outcomes: Admission, intensive care unit admission, mortality | 0.81% adverse events |
| Iqbal | USA | November 2020–January 2021 (21C) | Observational study | Outpatients | 284 | 149 | 68 | Bamlanivimab | Not treated | 14- and 30-day hospitalization and mortality | |
| Karr | USA | December 2020–January 2021 (21C) | Retrospective, cohort study | Outpatients, mild/moderate | 46 | 17 | 69 | 700 mg of bamlanivimab | Not treated | Clinical outcomes, Emergency department visits/admission | 10% adverse effects |
| Koehler | Germany | February 2021–April 2021 (Alpha and 20E) | Cross-sectional, retrospective study | Inpatients | 43 | 28 | 71.1 | 700 mg of LY-CoV555 (eight cases) and 2,400 mg of REGN-CoV-2 (casirivimab/imdevimab) (three cases). | Not treated | Clinical and therapeutical features and outcomes | 0% |
| Kumar | USA | November 2020–Jaunuary 2021 (21C) | Retrospective case-control study | Outpatients, mild/moderate | 376 | 193 | 64 | 700 mg of bamlanivimab | Not treated | Clinical outcomes | |
| Leavitt | USA | December 2020–January 2021 (21C) | Retrospective observational study | Outpatients | 279 | 149 | 63, 69 | Bamlanivimab | Not treated | 7-, 14-, and 28-day hospitalizations and mortality | None |
| Murillo | USA | November 2020–December 2020 (21C) | Retrospective study | Residents of nursing home, mild/moderate | 107 | 61 | 76 | 700 mg of bamlanivimab | Not treated | Hospitalizations, mortality | 5% adverse effects |
| Piccicacco | USA | November 2020–Jaunuary 2021 (21C) | Retrospective cohort study | Outpatients, mild/moderate | 400 | 207 | 63.2 | Bamlanivimab 700 mg (152 cases) or casirivimab/imdevimab 2,400 mg (48 cases) | Not treated | Clinical outcomes | 3.2–3.9% adverse effects |
| Rainwater-Lovett | USA | January 2021 (21C) | Retrospective study | Outpatients | 598 | 366 | 62.3 | Bamlanivimab | Not treated | 30-day medical visit | |

(Continued)

| Study | Country | Study period (prevalent strain) | Study design | Severity | Participants | Gender, female | Age (years) | Intervention | Control | Outcomes | Safety outcome |
|---|---|---|---|---|---|---|---|---|---|---|---|
| Rubin | USA | December 2020–February 2021 (21C) | Retrospective case-control study | Outpatients with BMI ≥35 or age ≥65 | 1,257 | 707 | 64.2 | Bamlanivimab | Not treated | 30-day hospitalizations or mortality | |
| Webb | USA | July 2020–January 2021 (21C) | Cohort study | Outpatients with risk score ≥7.5 points | 13,534 | 6,064 | 61 | Bamlanivimab 700 mg (479 cases) or casirivimab/ imdevimab 2,400 mg (115 cases) | Not treated | Clinical outcomes | 1.2% adverse events |

*2021*; *Iqbal et al., 2021*; *Leavitt et al., 2021*; *Murillo et al., 2022*; *Rainwater-Lovett et al., 2021*; *Rubin et al., 2021*).

### Meta-analysis of bamlanivimab treatment on primary and secondary outcomes

The quality assessment of the identified studies was summarized in Fig. 2. A low risks of bias in the randomized-controlled trials was observed in all domains of RoB-2. The concern of bias risk was noted in item 1 in the selection of all cohort studies. A total of 10 studies had potential risk of comparability, but the overall scores of the assessed studies were high (6 to 7 points). All of the studies were qualified to be enrolled in subsequent meta-analysis.

For assessing the risk of subsequent hospitalization, patients with BAM/ETE treatment had a significantly lower risk of hospitalization with moderate heterogeneity (18 trials, OR: 0.37, 95% CI [0.29–0.49], $I^2$: 69%; $p < 0.01$) (Fig. 3). A sensitivity test was conducted and the results remained unchanged by trials of individual studies. A funnel plot showed certain asymmetric distributions of the enrolled studies and was suggestive of publication bias (File S2). Another contour-enhanced funnel plot and Egger's test demonstrated the presence of publication bias (Files S3, S4; Egger's test, t = −2.66, $p = 0.0172$). Despite these results, BAM/ETE still had a significantly lower risk of subsequent hospitalization (eight trials, OR: 0.39, 95% CI [0.3–0.49], $I^2$: 32%; $p = 0.17$) after removing those studies with publication bias.

There was a significant reduction in mortality with BAM/ETE treatment and with low heterogeneity (15 trials, OR: 0.27, 95% CI [0.17–0.43], $I^2$: 0%; $p = 0.85$) (Fig. 4), however, some asymmetry was observed in the funnel plot (File S5). After removing studies with potential publication bias, the beneficial effects were similar (eight trials, OR: 0.24, 95% CI [0.11–0.51], $I^2$: 11%; $p = 0.35$). Furthermore, we conducted a subgroup analysis with BAM monotherapy which showed similar benefits (risk of subsequent hospitalization, 16 trials, OR: 0.43, 95% CI [0.34–0.54], $I^2$: 57%; $p = 0.01$) (Fig. 5). An additional funnel plot showed mild asymmetry and the Egger's test showed a low risk of publication bias (Files S6, S7; t = −2.2, $p = 0.0448$). BAM monotherapy was also shown to decrease subsequent mortality with low heterogeneity (14 trials, OR: 0.28, 95% CI [0.17–0.46], $I^2$: 0%; $p = 0.9$). Asymmetry was also observed in this funnel plot.

Two studies that investigated hospitalized patients showed no benefit of BAM treatments (*Koehler et al., 2021*; *Lundgren et al., 2021*). Adverse events ranged from 0–10%, including immediate hypersensitivity reactions and infusion related discomforts, which were tolerable. The hypersensitivity reactions that developed during infusion were reported as mild in severity and were without a dose-dependent relationship (*Gottlieb et al., 2021*; *Zhao et al., 2022*).

## DISCUSSION

Based on our systematic review and meta-analysis, non-hospitalized patients receiving BAM with or without ETE had a significantly lower risk of subsequent hospitalization and mortality (OR: 0.37 for hospitalization and 0.27 for mortality). This benefit was also

**(A) Randomized controlled trials**

| Rob-2 | Risk of bias domains | | | | | |
|---|---|---|---|---|---|---|
| Domains | D1 | D2 | D3 | D4 | D5 | Overall |
| BLAZE-1, monotherapy | + | + | + | + | + | + |
| BLAZE-1, combination | + | + | + | + | + | + |
| BLAZE-1, monotherapy vs combination | + | + | + | + | + | + |
| ACTIV-3 | + | + | + | + | + | + |

*D1: Randomization process; D2: Deviations from intended interventions; D3: Missing outcome data; D4: Measurement of the outcome; D5: Selection of the reported result.

** + : Low risk of bias

**(B) Cohort studies**

| Scale | Selection | | | | Comparability | Outcome | | | Overall scores |
|---|---|---|---|---|---|---|---|---|---|
| Item | 1 | 2 | 3 | 4 | 5 | 6 | 7 | 8 | |
| Alam | X | + | + | + | + | + | + | + | 7 |
| Bariola | X | + | + | + | + | + | + | + | 7 |
| Chilimuri | X | + | + | + | X | + | + | + | 6 |
| Cooper | X | + | + | + | + | + | + | + | 7 |
| Corwin | X | + | + | + | X | + | + | + | 6 |
| Destache | X | + | + | + | + | + | + | + | 7 |
| Ganesh | X | + | + | + | + | + | + | + | 7 |
| Iqbal | X | + | + | + | + | + | + | + | 7 |
| Karr | X | + | + | + | X | + | + | + | 6 |
| Koehler | X | + | + | + | X | + | + | + | 6 |
| Kumar | X | + | + | + | X | + | + | + | 6 |
| Leavitt | X | + | + | + | X | + | + | + | 6 |
| Murillo | X | + | + | + | X | + | + | + | 6 |
| Piccicacco | X | + | + | + | X | + | + | + | 6 |
| Rainwater-Lovett | X | + | + | + | + | + | + | + | 7 |
| Rubin | X | + | + | + | X | + | + | + | 6 |
| Webb | X | + | + | + | X | + | + | + | 6 |

*Item 1: Exposure truly representative of average; item 2: Selection of non-exposed from the same community; Item 3: Exposure ascertained by secure record or interview; Item 4: Demonstration of outcome of interest not present at the start of the study; Item 5: Study controls for other variables; Item 6: Follow up long enough for outcome to occur; Item 7: Complete follow up of all subjects accounted for; Item 8: Subject lost to follow up unlikely to introduce bias.

** + : Low risk  X : high risk of bias

**Figure 2 Quality assessment of included studies.**

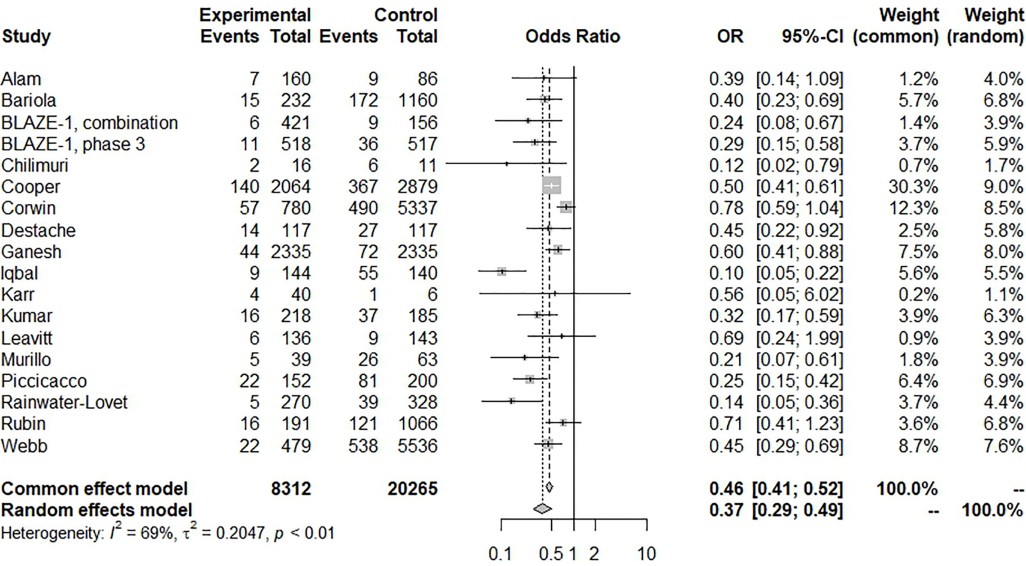

| Study | Experimental Events | Total | Control Events | Total | Odds Ratio | OR | 95%-CI | Weight (common) | Weight (random) |
|---|---|---|---|---|---|---|---|---|---|
| Alam | 7 | 160 | 9 | 86 | | 0.39 | [0.14; 1.09] | 1.2% | 4.0% |
| Bariola | 15 | 232 | 172 | 1160 | | 0.40 | [0.23; 0.69] | 5.7% | 6.8% |
| BLAZE-1, combination | 6 | 421 | 9 | 156 | | 0.24 | [0.08; 0.67] | 1.4% | 3.9% |
| BLAZE-1, phase 3 | 11 | 518 | 36 | 517 | | 0.29 | [0.15; 0.58] | 3.7% | 5.9% |
| Chilimuri | 2 | 16 | 6 | 11 | | 0.12 | [0.02; 0.79] | 0.7% | 1.7% |
| Cooper | 140 | 2064 | 367 | 2879 | | 0.50 | [0.41; 0.61] | 30.3% | 9.0% |
| Corwin | 57 | 780 | 490 | 5337 | | 0.78 | [0.59; 1.04] | 12.3% | 8.5% |
| Destache | 14 | 117 | 27 | 117 | | 0.45 | [0.22; 0.92] | 2.5% | 5.8% |
| Ganesh | 44 | 2335 | 72 | 2335 | | 0.60 | [0.41; 0.88] | 7.5% | 8.0% |
| Iqbal | 9 | 144 | 55 | 140 | | 0.10 | [0.05; 0.22] | 5.6% | 5.5% |
| Karr | 4 | 40 | 1 | 6 | | 0.56 | [0.05; 6.02] | 0.2% | 1.1% |
| Kumar | 16 | 218 | 37 | 185 | | 0.32 | [0.17; 0.59] | 3.9% | 6.3% |
| Leavitt | 6 | 136 | 9 | 143 | | 0.69 | [0.24; 1.99] | 0.9% | 3.9% |
| Murillo | 5 | 39 | 26 | 63 | | 0.21 | [0.07; 0.61] | 1.8% | 3.9% |
| Piccicacco | 22 | 152 | 81 | 200 | | 0.25 | [0.15; 0.42] | 6.4% | 6.9% |
| Rainwater-Lovet | 5 | 270 | 39 | 328 | | 0.14 | [0.05; 0.36] | 3.7% | 4.4% |
| Rubin | 16 | 191 | 121 | 1066 | | 0.71 | [0.41; 1.23] | 3.6% | 6.8% |
| Webb | 22 | 479 | 538 | 5536 | | 0.45 | [0.29; 0.69] | 8.7% | 7.6% |
| Common effect model | | 8312 | | 20265 | | 0.46 | [0.41; 0.52] | 100.0% | -- |
| Random effects model | | | | | | 0.37 | [0.29; 0.49] | -- | 100.0% |

Heterogeneity: $I^2 = 69\%$, $\tau^2 = 0.2047$, $p < 0.01$

0.1  0.5 1 2  10

**Figure 3 Forest plot of subsequent hospitalization of bamlanivimab ± etesevimab treatment.**

| Study | Experimental Events | Total | Control Events | Total | Odds Ratio | OR | 95%-CI | Weight (common) | Weight (random) |
|---|---|---|---|---|---|---|---|---|---|
| Alam | 5 | 160 | 9 | 86 | | 0.28 | [0.09; 0.85] | 11.2% | 18.0% |
| Bariola | 4 | 232 | 33 | 1160 | | 0.60 | [0.21; 1.71] | 10.6% | 20.8% |
| BLAZE-1, combination | 0 | 421 | 0 | 156 | | | | 0.0% | 0.0% |
| BLAZE-1, phase 3 | 0 | 518 | 10 | 517 | | 0.05 | [0.00; 0.80] | 10.3% | 2.8% |
| Chilimuri | 0 | 16 | 2 | 11 | | 0.12 | [0.00; 2.66] | 2.8% | 2.3% |
| Corwin | 1 | 780 | 35 | 5337 | | 0.19 | [0.03; 1.42] | 8.8% | 5.8% |
| Destache | 0 | 117 | 11 | 117 | | 0.04 | [0.00; 0.68] | 11.3% | 2.8% |
| Ganesh | 2 | 2335 | 8 | 2335 | | 0.25 | [0.05; 1.18] | 7.9% | 9.5% |
| Iqbal | 0 | 144 | 2 | 140 | | 0.19 | [0.01; 4.03] | 2.5% | 2.5% |
| Kumar | 1 | 218 | 4 | 185 | | 0.21 | [0.02; 1.88] | 4.2% | 4.7% |
| Leavitt | 0 | 136 | 0 | 143 | | | | 0.0% | 0.0% |
| Murillo | 4 | 39 | 18 | 63 | | 0.29 | [0.09; 0.92] | 12.2% | 16.7% |
| Piccicacco | 0 | 152 | 7 | 200 | | 0.08 | [0.00; 1.49] | 6.4% | 2.8% |
| Rubin | 1 | 191 | 10 | 1066 | | 0.56 | [0.07; 4.37] | 3.0% | 5.4% |
| Webb | 1 | 479 | 57 | 5536 | | 0.20 | [0.03; 1.46] | 8.9% | 5.8% |
| Common effect model | | 5938 | | 17052 | | 0.23 | [0.14; 0.37] | 100.0% | -- |
| Random effects model | | | | | | 0.27 | [0.17; 0.43] | -- | 100.0% |

Heterogeneity: $I^2 = 0\%$, $\tau^2 = 0$, $p = 0.85$

0.01  0.1  1  10  100

**Figure 4 Forest plot of subsequent mortality of bamlanivimab ± etesevimab treatment.**

observed in patients treated with BAM monotherapy (OR: 0.43 for hospitalization and 0.28 for mortality). The adverse events of both of these treatments were tolerable. A single BAM/ETE infusion provided in an emergency department or infusion center was a reasonable and feasible treatment for non-hospitalized patients. This treatment led to decreased subsequent hospitalization and mortality and preserved medical resources during the study period when the Alpha variant was the predominant circulating strain.

Although vaccination plays a crucial role in the battle against COVID-19, effective anti-viral agents are indispensable (*Chiu et al., 2021*; *Chiu et al., 2022*). The stage and severity of the disease may affect the strategy and effects of treatment. There are two

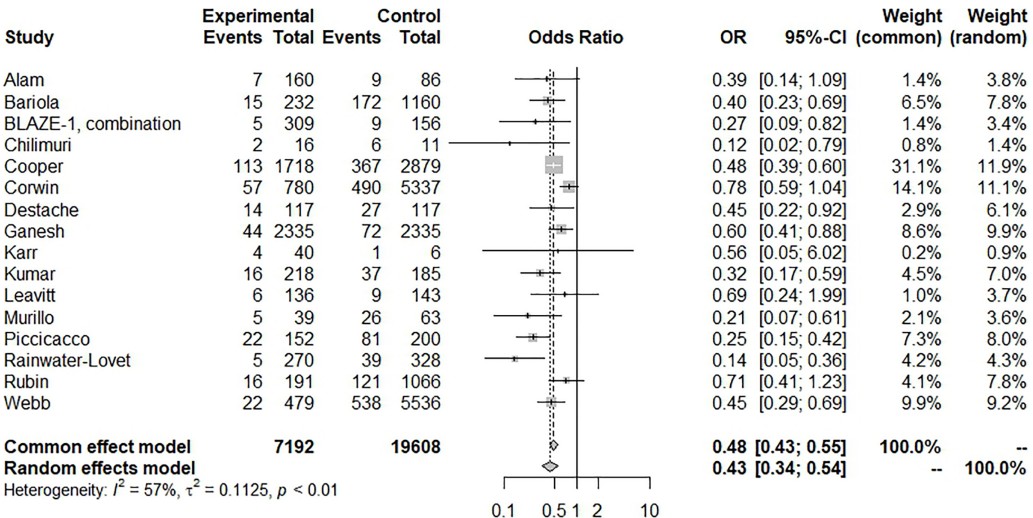

**Figure 5 Forest plot of subsequent hospitalization of bamlanivimab monotherapy.**

significantly different stages in the pathophysiology of COVID-19: virus replication and subsequent inflammation. Monoclonal antibodies, convalescent plasma, and antiviral agents block viral entry and abrogate viral replication for patients with no symptoms or only mild illness, as seen in the early stage (*Gandhi, Lynch & del Rio, 2020*). Corticosteroid and immunomodulatory agents may be applied to mitigate the inflammatory cascade for patients with moderate or severe illness in the later stages of the disease (*Hariyanto, Hardyson & Kurniawan, 2021*). Therefore, the use of BAM/ETE on hospitalized patients with moderate or severe infection is ineffective (*Borrok et al., 2018*). BAM/ETE targets the spike protein of SARS-CoV-2, and early BAM/ETE treatment will block the virus's entry, which is beneficial for the successful treatment of COVID-19 (*Cohen, 2021*; *Jones et al., 2020*). However, BAM/ETE is not advantageous for hospitalized patients. For these patients, viral replication is robust and a high viral load and inflammatory cytokines are typically observed. Providing passive immunity to inhibit the viral spike protein dos not modify the subsequent inflammatory processes. Similar findings were reported in remdesivir, an RNA-dependent RNA polymerase inhibitor. The use of remdesivir has been shown to be beneficial to patients receiving oxygen; however, the outcomes were not significantly different in ventilated patients (*Beigel et al., 2020*). Therefore, the treatment timing is imperative for BAM/ETE in the clinical setting. For patients with mild illness and risk factors for severe infection, including older age, obesity, and other underlying systemic diseases, timely BAM/ETE administration is crucial for successful treatment.

In addition to disease timing and severity, rapidly mutating variants of the virus affect the effects of pharmacologic treatment, and drug resistance is the main barrier for the clinical application of monoclonal antibodies (*Tuccori et al., 2021*). BAM is not effective against the variants B.1.351., P.1., B.1.6.7.2 (Delta), and B.1.1.529 (Omicron). Pseudovirus neutralizing antibody testing showed that BAM was effective against variants with the N501Y mutation (B.1.1.7., Alpha), but there was a significant decrease in sensitivity against

variants with the E484K mutation (B.1.351, P.1, B.1.526, and B.1.617), and the L452R mutation (B.1.427 and B1.429) (*Hoffmann et al., 2021*). Resistance was also confirmed by a neutralizing antibody test of sera and a significantly-reduced sensitivity to the B1.6.7.2 (Delta) variant was observed (*Planas et al., 2021*). Mutation in the N-terminal domain and the receptor binding domain of the spike protein may be responsible for the observed immunological evasion and escape. However, ETE, casirivimab, and imdevimab remained sensitive to the Delta variant but not Omicron. As Delta and Omicron variants become the predominant strains in many countries, monoclonal antibody treatment with BAM/ETE should be avoided. Our study found a significantly reduced risk of hospitalization and mortality in both BAM monotherapy and a combination therapy with BAM and ETE. The prevalent strains and variants of interest were 21C (Epsilon), 20E (EU1). However, the variant of concern, B.1.1.7 (Alpha), was prevalent in the included studies. The clinical application of BAM/ETE is limited by the prevalence of rapidly mutating variants. Moreover, the widespread use of monoclonal antibodies may accelerate the emergence of COVID mutants (*Focosi et al., 2022*) since the monoclonal antibody targets single epitopes and the spike protein escape mutation may occur under specific selective pressure. For example, the simultaneous resistance to BAM and ETE with the spike mutation Q943R was reported in a patient given BAM/ETE treatment (*Focosi et al., 2021*). Therefore, genomic surveillance is recommended, especially for patients with poor responses to monoclonal antibody treatment.

The COVID-19 pandemic is novel and there is a battle between viral mutations and available treatments. Several monoclonal antibodies have been developed, similar to BAM/ETE, including casirivimab plus imdevimab, sotrovimab, tixagevimab plus cilgavimab, and bebtelovimab. The use of BAM/ETE and casirivimab/imdevimab was halted due to the resistance of the Omicron mutants, which have been the predominant strains worldwide since late 2021. Sotrovimab is a medication that was initially identified in a patient with SARS-CoV infection. It was shown to target an epitope conserved in both SARS-CoV and SARS-CoV-2. Sotrovimab is effective against the Omicron variant and has been authorized to treat mild-to-moderate COVID-19 in non-hospitalized patients at high risk for progressing to severe disease. Bebtelovimab is a broadly neutralizing monoclonal antibody targeting relatively conserved epitopes. It is effective against the current variants, including those of Omicron (*Dougan et al., 2022*; *Westendorf et al., 2022*). This treatment has received emergency use authorization from the Food and Drug Administration to treat mild to moderate COVID-19 patients (*US Food & Drug Administration, 2020*).

Furthermore, like the use of the highly active antiretroviral therapy to overcome the resistance of human immunodeficiency virus, cocktail therapies with different kinds monoclonal antibodies have been developed to treat COVID-19 variants. These cocktails include combinations of bebtelovimab, BAM, and ETE (*Dougan et al., 2022*). Additional cocktail combinations using monoclonal antibodies and antiviral agents have also been studies. BAM/ETE has been shown to be ineffective against the Delta and Omicron variants and its clinical application is limited at present. More options are required when facing rapidly mutating viruses. Many older drugs have been repurposed in an attempt to treat COVID-19 in its different stages and severity. The use of BAM/ETE is a reminder of

the importance of genomic surveillance it may be repurposed as a component of a cocktail regimen for treating future variants.

Monoclonal antibody treatment is an attractive treatment option because it may be provided as a single infusion without hospitalization. For example, peramivir is an anti-influenza viral agent that has shown success with a single dose that can be provided in an outpatient setting (*Hsieh et al., 2021*). A decrease in subsequent hospitalizations is valuable for its ability to preserve vital medical capacity necessary during the pandemic. Furthermore, BAM may also used for prophylaxis after close contact with an infected person (*Cohen et al., 2021*). Residents in nursing and assisted living facilities are at higher risk of contracting the disease during a COVID-19 outbreak. BAM prophylaxis could significantly decrease the risk of COVID-19 infection in residents and staff (BAM *vs.* placebo: 8.5% *vs.* 15.2%; odds ratio, 0.43 95% CI [0.28–0.68]) and its administration is feasible (*Dale et al., 2021*). However, the COVID variant will affect the prophylactic effects of BAM and more evidence is required to determine its benefits.

Our study is subject to certain limitations. First, the virus has been shown to mutate rapidly and the prevalent strain at present is different from those of the study periods. Although our study shows the significant benefits of early BAM/ETE treatment, *in vitro* studies showed a decreased sensitivity of BAM/ETE against some variants of concern. Most hospitals do not have the ability to test for specific COVID variants, thus the use of BAM/ETE is limited. BAM/ETE is ineffective against variants of concern, including Delta and Omicron. However, mutation occurs rapidly and unexpectedly, so repurposing old drugs may provide a potential treatment option when facing a new variant. Second, funnel plots and Egger's tests indicated the presence of publication bias. Studies with negative or inconclusive results may not be published. Finally, the benefits of BAM/ETE are obvious in non-hospitalized patients, but the criteria of hospitalization may vary in different health care units. Further objective saturation or laboratory testing may assist primary care physicians to identify the patients who would be most appropriate for BAM/ETE treatment.

## CONCLUSIONS

In conclusion, we found a significant reduction of subsequent hospitalization and mortality in non-hospitalized patients treated with BAM, with or without ETE treatment (OR: 0.37 for subsequent hospitalization, moderate heterogeneity, and 0.27 for mortality, low heterogeneity). Adverse events were not frequent and were tolerable to the 28,577 patients identified in our systematic review and meta-analysis. A single infusion with BAM with or without ETE in an outpatient setting was feasible for patients with risk factors and mild illness. This has been shown to be a reasonable treatment strategy to decrease subsequent hospitalization and to preserve medical capacity which ensure timely and adequate medical resources for patients with severe COVID-19. However, the effectiveness of BAM/ETE is challenged by the rapidly emerging variants of SARS-CoV-2.
The application course of BAM/ETE is a reminder of the importance of genomic surveillance, and the repurposed use of BAM/ETE may provide a potential treatment regimen in treating future variants.

## ACKNOWLEDGEMENTS

We appreciate the efforts of those who have worked to combat COVID-19.

### Funding

The authors received no funding for this work.

### Competing Interests

The authors declare that they have no competing interests.

### Author Contributions

- Yu-Lin Tai conceived and designed the experiments, analyzed the data, prepared figures and/or tables, and approved the final draft.
- Ming-Dar Lee conceived and designed the experiments, performed the experiments, analyzed the data, prepared figures and/or tables, and approved the final draft.
- Hsin Chi conceived and designed the experiments, prepared figures and/or tables, and approved the final draft.
- Nan-Chang Chiu performed the experiments, authored or reviewed drafts of the article, and approved the final draft.
- Wei-Te Lei performed the experiments, analyzed the data, authored or reviewed drafts of the article, and approved the final draft.
- Shun-Long Weng performed the experiments, authored or reviewed drafts of the article, and approved the final draft.
- Lawrence Yu-Min Liu analyzed the data, authored or reviewed drafts of the article, and approved the final draft.
- Chung-Chu Chen performed the experiments, authored or reviewed drafts of the article, and approved the final draft.
- Shih-Yu Huang performed the experiments, authored or reviewed drafts of the article, and approved the final draft.
- Ya-Ning Huang performed the experiments, authored or reviewed drafts of the article, and approved the final draft.
- Chien-Yu Lin conceived and designed the experiments, performed the experiments, analyzed the data, prepared figures and/or tables, authored or reviewed drafts of the article, and approved the final draft.

### Data Availability

The raw data is available in the Supplemental Files.

### Supplemental Information

Supplemental information for this article can be found online at http://dx.doi.org/10.7717/peerj.15344#supplemental-information.

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
