# Peer review of "Effects of bamlanivimab alone or in combination with etesevimab on subsequent hospitalization and mortality in outpatients with COVID-19: a systematic review and meta-analysis"

_PeerJ, doi:10.7717/peerj.15344_

## Round 0.1 · original submission · Major Revisions

Thank you for submitting your manuscript. After careful consideration, we feel that it has merit but does not fully meet Journal’s publication criteria as it currently stands. Therefore, we invite you to submit a revised version of the manuscript that addresses the points raised during the review process.

Reviewer 1 ·

Basic reporting

No comment. Relevant results and it did answered the research question. Writing was good however it could be improved.

Experimental design

No comment, used methodology is strong and valid.

Validity of the findings

Findings are relevant, interesting to share.

Additional comments

This paper was well presented and goodly wrote, the methodology was strong and correct with no red flags nor big errors, the conclusion responded the research question, results are interesting and can be used for further studies.

Reviewer 2 ·

Basic reporting

1. Need to improve the language of the manuscript. Sufficient background for the study has been given. Results are relevant to their intended objectives

Experimental design

2. Research question and PICO format should be added, for better understanding of the characteristics of the studies included.
3. Lines 87-92 does not match well with the subsequent lines. May need to rearrange them at appropriate place.
4. Line 174- “Most studies were conducted in the US (one),”- Check the content
5. Under study selection, its mentioned “randomized-controlled studies or cohort studies”. Yet, Koehler et al and Kumar et al are Cross sectional and case control studies, respectively. Kindly clarify on this contradiction
6. Authors may add the ethical statement (or its non-applicability) of the research

Validity of the findings

7. The search date is more than 16 months old. Authors need to do an updated search for latest date, and add new studies if any. This is of importance considering fact that emerging variants affects the efficacy of the ab.

Reviewer 3 ·

Basic reporting

Basic reporting
1. The English is overall acceptable. Some tense correction is needed – I have annotated some of these in the accompanying pdf. The authors are requested to go through the entire manuscript with a fluent speaker.
2. Intro and background are relevant and well referenced
a. Case fatality rate needs referencing (line 66)
b. What is the meaning of co-existing policy? (Line 88)
3. Figures are relevant, high quality, well labelled and described
a. Yes
4. Raw data supplied
a. Yes

Experimental design

Experimental Design
1. Does the manuscript match the scope of the journal?
a. Yes
2. Research Question well defined?
a. Yes, although using a PICOT format will add clarity.
3. Rigorous investigation performed to a high technical and ethical standard?
a. These have been followed
4. Methods:
a. There is a clear disparity in the list of databases included in this meta-analysis: lines 123, 41 and 42 – This is quite concerning
b. Two third authors breaking a tie seems unusual
c. Instead of using terms as “small study bias”, “publication bias” might be a better term (Line 160)
5. Results section
a. What is being mean by “(one)” – line 174 needs rephrasing
b. Authors should probably analyze and provide a summary treatment effect after removing trials with publication bias (Lines 194- 98)
c. I would not really call anaphylaxis as a tolerable side effect. If data is available in the original manuscript, please provide the frequency of anaphylaxis observed.

Validity of the findings

Validity of the findings
1. Benefit to literature clearly stated
a. Yes
2. All underlying data provided?
a. Yes
3. Robust data?
a. Yes
4. Statistically sound?
a. Yes. Although an analysis after removing studies contributing to publication bias would be quite useful.
5. Conclusions are well stated, linked to original research question and limited to supporting results?
a. The limitations and conclusions are well stated
b. A note on the moderate heterogeneity in the primary outcome is needed under the “Conclusion” section.

Additional comments

English needs minor improvement, else a well written manuscript. Please not an annotated pdf has also been provided

Annotated reviews are not available for download in order to protect the identity of reviewers who chose to remain anonymous.

---

## Round 0.2 · Minor Revisions

The ongoing COVID-19 pandemic has resulted in an unprecedented global health crisis, leading to numerous medical advancements and research initiatives to address the needs of affected individuals. One such area of research has been the exploration of the efficacy of bamlanivimab and etesevimab in preventing subsequent hospitalization and mortality in outpatients with COVID-19. The results of this systematic review and meta-analysis provide valuable insights into the topic.

Please address the remaining reviewer comments.

Reviewer 2 ·

Basic reporting

The authors have added a certificate of professional services to improve the language. However, there is still scope for improvement of the language

Experimental design

no comment

Validity of the findings

All data to validate their conclusion are provided

Reviewer 3 ·

Basic reporting

The authors have certainly improved the English. There is still room for improvement.

Experimental design

Since the authors have included more trial as part of an updated search table 1, figure 1 and 2 should be updated. The ROB and quality assessments need to be done for the newly included studies as well. This may not change the results or interpretation, however systematic reviews and meta-analyses need to maintain certain standards.

Validity of the findings

The findings are certainly valid

Additional comments

I would recommend rewriting the results with the updated search rather than including the results of the updated search as a supplement. The current presentation makes the immense effort put in by the authors as haphazard.

---

## Round 0.3 · accepted · Accept

The authors have addressed all of the reviewers' comments. The manuscript may be accepted as per the Journal Policy.